# EcoXplain: An Interpretable Causal-Augmented Framework for Macroeconomic Forecasting

## Abstract

The urgency of policy response demands not only accurate predictions but also a deep understanding of causal mechanisms, which has become increasingly challenging as structural relationships between economic variables evolve over time. Existing machine learning approaches often function as black boxes, achieving high predictive accuracy but providing little interpretability, while traditional structural models struggle to adapt to fast-moving causal dynamics. To address these challenges, we propose **EcoXplain**[1], an interpretable dynamic causality augmented spatio-temporal graph neural network architecture specifically designed for low-frequency macroeconomic data. EcoXplain combines interpretable white-box decomposition with a Dynamic Spatio-Temporal Graph Neural Network (DSTGNN), which integrates short-term inferred dynamic causal relationships with prediction-driven adaptive adjacency matrices that capture evolving relationships between macroeconomic variables. Empirical evaluations on datasets from four major economies (China, Japan, the US, and the UK) show that EcoXplain significantly outperforms both white-box methods and state-of-the-art black-box machine learning baselines, reducing forecasting error (MAPE) by up to **88.98%** and **68.07%**, respectively. Beyond predictive gains, EcoXplain uncovers meaningful causal pathways that provide policymakers and economists with a deeper understanding of how economic forces interact, offering valuable guidance for timely and effective decision-making.

## 1 Introduction

When faced with economic crises such as deflation and rising unemployment, policymakers must rapidly deploy targeted interventions through interest rate adjustments or government spending to immediately counteract economic contraction and alleviate unemployment pressures (Blanchard et al., 2009). However, as traditional forecasting models increasingly fail under recent changes in growth momentum (Ng, 2021) and new economic dynamics emerge, policymakers face the daunting challenge of implementing potentially untested policy measures to prevent economic and social damage. The urgency of policy response therefore demands not only accurate predictions but also deep understanding of causal mechanisms. Policymakers and economists must identify key influencing factors, construct the transmission pathways of causal relationships, and understand their dependencies to make informed decisions that can effectively address economic challenges (BLANCHARD et al., 2010).

Although a rich body of time-series models has long been applied to macroeconomic forecasting, most of these studies focus on a single ML model and examine only limited forecasting horizons. While these models may achieve superior performance in forecasting competitions, they fail to provide insights into how macroeconomic variables are interconnected and mutually influenced (Coulombe et al., 2020). Essentially, the black box remains closed, rendering these approaches largely unusable for economists and policymakers who require interpretable mechanisms to inform decision-making. At the same time, the relationships between economic variables exhibit unprecedented volatility and complexity, with causal dependencies that continuously evolve over time. More

---

[1]Our code is publicly available at `https://anonymous.4open.science/r/EcoXplain-2A5C`.

recently, spatio-temporal graph methods have attracted attention, but their application to macroeconomic datasets has proved disappointing. Unlike traffic or mobility data, macroeconomic systems lack clear ground-truth relational structures, and the data are typically low-frequency and small in scale, making it difficult for conventional methods to extract robust causal dependencies.

Simultaneously, the digital economy era provides unprecedented opportunities for macroeconomic modeling, as sophisticated economic behaviors can now be systematically recorded and analyzed (Cheng et al., 2021). The availability of abundant exogenous data enables the modeling of complex environments and motivates the development of innovative data-driven methods to improve forecasting accuracy and reveal evolving economic relationships.

To address these issues, we propose a novel model architecture that synergistically combines economic-based white-box decomposition for interpretable temporal modeling with a Dynamic Spatio-Temporal Graph Neural Network (DSTGNN) that models evolving causal dependencies. We combine the short-term causal dependencies inferred within each sliding window with the long-term causal relationships distilled from the predictive process, thereby capturing both transient dynamics and more persistent structural linkages. Beyond achieving accurate forecasting performance, we emphasize the importance of understanding how causality between factors evolves over time, since such knowledge can fundamentally change perspectives on economic policy formulation and implementation.

The main contributions of this paper are as follows:

- We propose **EcoXplain**, the first hybrid framework for macroeconomic forecasting that combines interpretable white-box decomposition with causal-augmented dynamic spatio-temporal graph modeling. This design directly enables both accurate predictions and interpretable causal mechanisms, which is essential for economic policy analysis.

- We develop a Dynamic Spatio-Temporal Graph Neural Network (DSTGNN) that integrates short-term inferred causal relationships with long-term prediction-driven dependencies. The DSTGNN captures both transient dynamics and hidden interdependencies, thereby enhancing forecasting accuracy while improving interpretability.

- We conduct extensive experiments on macroeconomic datasets from four major economies (China, Japan, the US, and the UK). Results demonstrate that EcoXplain consistently outperforms classical statistical models and state-of-the-art machine learning baselines, reducing forecasting error (MAPE) by up to **88.98%** compared with white-box methods and **68.07%** compared with state-of-the-art black-box machine learning baselines, while simultaneously uncovering evolving causal structures between economic variables.

## 2 RELATED WORKS

**White-box Models.** Interpretable (white-box) models are widely regarded as the mainstream solution for macroeconomic forecasting and can be broadly categorized into data-driven and non-data-driven approaches. Classical economics-based methods, such as Bayesian Vector AutoRegressive Integrated Moving Average(ARIMA) (Box & Jenkins, 1970; Box et al., 2015), are non-data-driven and therefore lack adaptability to dynamic, real-time environments. In contrast, data-driven methods such as N-BEATS (Oreshkin et al., 2020), a state-of-the-art interpretable neural forecasting model, have achieved strong performance in macroeconomic prediction (Wang et al., 2022). Beyond that, N-BEATSx (Olivares et al., 2022) extends the original framework by incorporating external covariates into the forecasting process.

**Black-box Models.** Alongside white-box approaches, black-box models, particularly Transformer-based architectures, have recently achieved significant breakthroughs in time-series forecasting. Informer (Zhou et al., 2021) introduces ProbSparse self-attention and a generative decoder for long-sequence prediction. Pyraformer (Liu et al., 2022) captures multi-resolution representations with a pyramidal attention module. Autoformer (Wu et al., 2021) leverages an auto-correlation mechanism with decomposition blocks, while FEDformer (Zhou et al., 2022) enhances forecasting with Fourier and Wavelet components. More recently, iTransformer (Liu et al., 2023) reformulates tokenization by treating variables as tokens and temporal positions as features. In parallel, UniTST (Shan et al., 2024) unifies inter- and intra-series modeling with patch-wise

tokens and lightweight attention. These Transformer-based methods demonstrate remarkable accuracy, but their black-box nature poses challenges for interpretability and economic insight.

**Spatio-Temporal Graph Neural Networks.** Beyond temporal models, spatio-temporal graph neural networks (STGNNs) have become a major frontier in time-series forecasting. Early works such as DCRNN (Li et al., 2018), STGCN (Yu et al., 2018), and Graph WaveNet (Wu et al., 2019) captured spatial correlations through diffusion, gated convolutions, or adaptive adjacency. Later models like AGCRN (Bai et al., 2020) and DGCRN (Li et al., 2021) introduced adaptive and dynamic graph construction, while DVGNN (Liang et al., 2023) further integrated variational inference to learn probabilistic time-varying causal graphs. Overall, STGNNs have evolved from static graph convolutional models to dynamic, causality-aware, and Transformer-augmented frameworks, which excel in forecasting domains such as traffic and sensors. In macroeconomic forecasting, however, they have so far delivered limited gains: most architectures are tuned for large-scale, high-frequency data while macro series are sparse and low-frequency, and the induced graph structures remain shallow and insufficient to capture cross-sector economic causal links.

## 3 METHODOLOGY

### 3.1 PROBLEM FORMULATION

The EcoXplain framework is designed with a decomposable structure that addresses the complexity of multivariate economic forecasting by integrating both temporal dependencies and dynamic causal relationships. We consider a multivariate forecasting problem with two types of inputs: the target series representing the primary economic indicators to be predicted, and exogenous covariates capturing external influential factors.

Let $\mathbf{Y}_{1:T} = \{\mathbf{Y}_1, \mathbf{Y}_2, \ldots, \mathbf{Y}_T\}$, where $\mathbf{Y}_t \in \mathbb{R}^K$ denotes the target series at time $t$ ($K$ represents the number of target variables). Similarly, let $\mathbf{X}_{1:T} = \{\mathbf{X}_1, \mathbf{X}_2, \ldots, \mathbf{X}_T\}$, where $\mathbf{X}_t \in \mathbb{R}^D$ represents the exogenous covariates ($D$ denotes the number of external variables).

The primary objective is to predict the future horizon of the target series:

$$\hat{\mathbf{Y}}_{T+1:T+H} = f_\Theta\big(\mathbf{Y}_{1:T}, \mathbf{X}_{1:T}, \mathcal{G}_{1:T}\big), \quad \mathcal{G}_{1:T} = \{\mathbf{A}^{(1)}, \mathbf{A}^{(2)}, \ldots, \mathbf{A}^{(T)}\}, \quad \mathbf{A}^{(t)} \in \mathbb{R}^{N \times N}, \quad (1)$$

where $f_\Theta(\cdot)$ represents the spatio-temporal forecasting model parameterized by $\Theta$, $H$ is the forecasting horizon, and $\mathcal{G}_{1:T}$ denotes the dynamic graph sequence that characterizes temporal causal dependencies among the $N = K + D$ variables.

The proposed pipeline synergistically combines a white-box temporal forecaster with a dynamic spatio-temporal graph neural network, as shown in Figure 1. This hybrid architecture leverages the interpretability of hierarchical decomposition while capturing complex spatio-temporal dependencies through dynamic causal graphs.

### 3.2 WHITE-BOX DECOMPOSITION

Inspired by the N-BEATS architecture (Oreshkin et al., 2020), the basic building unit of our pipeline is a block, where each white-box block possesses a unique basis vector $\mathbf{v}_k$ for the $k$-th block and generates two components: a forecast component $\hat{\mathbf{y}}_k^f$ of length $H$ for future predictions, and a backcast component $\hat{\mathbf{y}}_k^b$ of length $T$ for historical reconstruction. The backcast component serves to fit existing economic data patterns, while the forecast component provides future-oriented predictions.

The computation process within each white-box block follows:

$$\hat{\mathbf{y}}_k^q = \sum_{i=1}^{|\boldsymbol{\theta}_k^q|} \theta_{k,i}^q v_{k,i}^q, \quad \boldsymbol{\theta}_k^q = \text{Linear}^q(\text{FC}_k(\mathbf{x}_{k-1}^b)), \quad (2)$$

where $q \in \{f, b\}$ indicates forecast or backcast, $\mathbf{x}_{k-1}^b$ represents the input to the current block (residuals from the previous block), and $\boldsymbol{\theta}_k^q$ denotes the expansion coefficients that generate the respective output components.

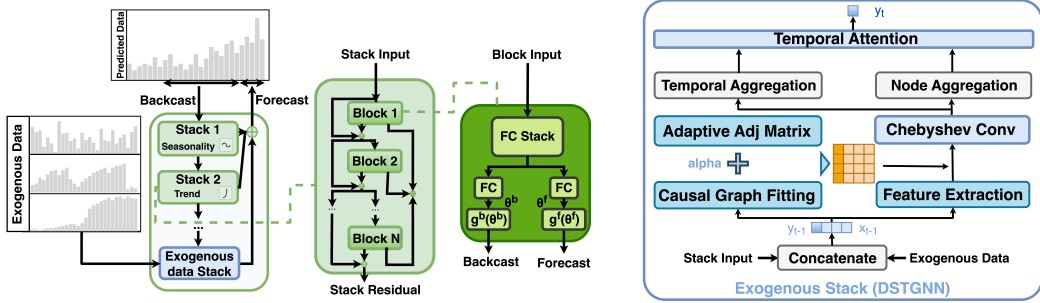

Figure 1: **EcoXplain Framework:** The left panel presents the hierarchical structure of the model, while the right panel highlights the DSTGNN module, which serves as the exogenous data stack within the overall architecture.

The hierarchical structure operates through residual subtraction stacking:

$$\mathbf{y}_k^b = \mathbf{y}_{k-1}^b - \hat{\mathbf{y}}_{k-1}^b, \quad \hat{\mathbf{y}}^f = \sum_{k=1}^{N_{\text{blocks}}} \hat{\mathbf{y}}_k^f, \tag{3}$$

where the backcast components are subtracted from the input as residuals for the next block, while forecast components are summed across all blocks to produce the final prediction.

Given the inherent regularities in macroeconomic dynamics, the white-box part of our framework is designed to fully utilize economic structural knowledge in our prediction stage, which emphasizes the representation of cyclical fluctuations and long-term trends as the following functions.

$$\hat{\mathbf{y}}_k^{\text{trend}} = \sum_{i=0}^{H-1} \theta_{k,i}^{\text{trend}} t_i \tag{4}$$

$$\mathbf{y}_k^{\text{seasonality}} = \sum_{i=0}^{\lfloor H/2 \rfloor - 1} \theta_{k,i}^{\text{seasonality}} \cos(2\pi it) + \sum_{i=1}^{\lfloor H/2 \rfloor} \theta_{k,i}^{\text{seasonality}} \sin(2\pi it) \tag{5}$$

Macroeconomic information related to economic cycles and trends are extracted upfront by constructing the constrained stacks T and S. The following module then focuses solely on the residual data after elimination by the T and S stacks.

### 3.3 GRANGER CAUSALITY DISCOVERY

**Granger Causality Foundation.** Our approach builds upon the concept of Granger causality to establish temporal causal relationships. To accommodate the nonlinear dynamics prevalent in economic systems, we extend this to the nonlinear case where:

$$f_j(\mathbf{X}_{t-\tau:t-1,1}, \ldots, \mathbf{X}'_{t-\tau:t-1,i}, \ldots, \mathbf{X}_{t-\tau:t-1,N}) \neq$$
$$f_j(\mathbf{X}_{t-\tau:t-1,1}, \ldots, \mathbf{X}_{t-\tau:t-1,i}, \ldots, \mathbf{X}_{t-\tau:t-1,N}), \tag{6}$$

for some $\mathbf{X}'_{t-\tau:t-1,i} \neq \mathbf{X}_{t-\tau:t-1,i}$, where $f_j(\cdot)$ represents a nonlinear predictive mapping. This condition indicates that the historical values of variable $i$ have a substantive impact on the prediction of variable $j$.

**Dynamic Causal Graph Fitting.** Real-world economic and financial systems exhibit complex, time-varying causal relationships that cannot be adequately captured by static graph structures. While some domains possess well-established causal relationships, economic systems often involve latent causal dependencies between variables. To capture the time-varying nature of causal dependencies in economic systems, we extend the data imputation and causal graph fitting framework to construct time-varying causal graphs that capture the dynamic nature of economic dependencies in a sliding window:

$$\mathbf{A}_{\text{dynamic}}^{(t)} = \langle X_{t-w:t}, \{M_\tau^{(t)}\}_{\tau=0}^{\tau_{max}} \rangle \tag{7}$$

In this formulation, $X_{t-w:t}$ represents the set of variables within the sliding window. The causal strength from variable $i$ to variable $j$ at time $t$ is quantified by $m_{\tau,ij}^{(t)} \in M_\tau^{(t)}$, and each dynamic adjacency entry $a_{ij}^{(t)}$ is inferred by:

$$a_{i,j}^{(t)} = \max_{\tau \in \{1,\ldots,\tau_{max}\}} (m_{1,ij}^{(t)}, \ldots, m_{\tau_{max},ij}^{(t)}) \tag{8}$$

Form which we know that if $a_{i,j}^{(t)}$ is penalized to zero, time-series $i$ does not Granger cause time-series $j$ at time $t$. In this work, we use $w$ to control the length of sliding windows in order to better illustrate the short-term interplay between observations (see Appendix A.3). The discovered dynamic causal graphs serve as our starting point for causal graph construction.

### 3.4 DYNAMIC SPATIO-TEMPORAL GRAPH NEURAL NETWORK (DSTGNN)

The Dynamic Spatio-Temporal Graph Neural Network (DSTGNN) operates as the final enhancement block, processing residual information from white-box forecasting through a sophisticated combination of Chebyshev graph convolution and temporal attention mechanisms and enhancing predictions with integrated dynamic causal matrices, as show in Figure 1.

**Hybrid Causal Matrices Construction.** For each batch sample of tne whole time step $T$, the DSTGNN operates on the processed residual data $\mathbf{R}_t \in \mathbb{R}^{B \times C \times N \times L}$ after white-box stacks, where $B$ denotes batch size, $C$ represents the number of channels, $N$ is the number of variables, and $L$ is the time window.

DSTGNN employs a GraphWaveNet-inspired architecture with adaptive adjacency learning at each prediction time point. Unlike causal discovery period, this matrix do not run data imputation period and only focus on the final prediction stage. The adaptive adjacency matrix is computed as:

$$\mathbf{A}_{\text{adapt}}^{(T)} = \text{softmax}\left( \text{ReLU}\left( \mathbf{E}_1^{(T)} \mathbf{E}_2^{(T)\top} \right) \right), \tag{9}$$

with $\mathbf{E}_1^{(T)}, \mathbf{E}_2^{(T)} \in \mathbb{R}^{N \times d}$ being learnable node embedding matrices of dimension $d$, from which the adaptive adjacency at prediction step $T$ is constructed. This design greatly reduces memory usage and enables the incorporation of a larger number of variables into the prediction stage.

The final adjacency matrix combines multiple sources of structural information:

$$\mathbf{A}^{(t \to T)} = \alpha \mathbf{A}_{\text{adapt}}^{(T)} + (1-\alpha) \mathbf{A}_{\text{dynamic}}^{(t)}, \tag{10}$$

where $\alpha \in [0,1]$ is a weighting parameter that balances between learned adaptive relationships and discovered dynamic causal relationships. Here, $\mathbf{A}_{\text{dynamic}}^{(t)}$ denotes the time-specific slice of the dynamic causal graph, while $\mathbf{A}_{\text{adapt}}^{(T)}$ is re-estimated for each prediction horizon and adaptively captures the evolving dependencies between target variables and exogenous covariates across time.

**Chebyshev Spatio-Temporal Convolution.** To effectively capture spatial dependencies, we adopt a Chebyshev polynomial approximation of graph convolutions, which avoids expensive eigendecomposition while retaining expressive power. Specifically, we recursively construct the Chebyshev basis:

$$\mathbf{L}_0 = \mathbf{I}_N, \quad \mathbf{L}_1 = \mathbf{A}^{(t \to T)}, \quad \mathbf{L}_k = 2\mathbf{A}^{(t \to T)}\mathbf{L}_{k-1} - \mathbf{L}_{k-2}, \quad k = 2, \ldots, K, \tag{11}$$

where $\mathbf{A}^{(t \to T)}$ is the learned dynamic adjacency encoding causal relationships. Each $\mathbf{L}_k$ captures spatial relationships at different hop distances: $\mathbf{L}_0$ represents self-connections, $\mathbf{L}_1$ captures direct causal influences, and higher-order terms $\mathbf{L}_k$ model multi-hop dependencies through intermediate variables.

The constructed basis is then aggregated through graph convolution:

$$\mathbf{H}_{\text{graph}} = \text{einsum}(\mathbf{R}_t, \text{stack}([\mathbf{L}_0, \ldots, \mathbf{L}_{K-1}])), \tag{12}$$

which projects node embeddings into a spatially enriched representation. A subsequent temporal convolution further extracts local temporal patterns:

$$\mathbf{Z}_{\text{spatial}} = \text{Conv2D}(\mathbf{H}_{\text{graph}}; \mathbf{W}_{\text{conv}}). \tag{13}$$

**Temporal Attention Enhancement.** We introduce a dual-branch temporal attention mechanism that adaptively learns cross-time dependencies. Two permutation-based convolutions,

$$\mathbf{F}_1 = \text{Conv2D}(\mathbf{R}_t^{\text{perm}_1}; \mathbf{W}_1), \quad \mathbf{F}_2 = \text{Conv2D}(\mathbf{R}_t^{\text{perm}_2}; \mathbf{W}2), \tag{14}$$

encode complementary temporal and spatial perspectives separately. The resulting features are then combined through an attention module, where the attention coefficients are computed as:

$$\mathbf{A}_{\text{coeff}} = \text{softmax}(\mathbf{v} \cdot \sigma(\mathbf{F}_1 \mathbf{W} \mathbf{F}_2^T + \mathbf{b}) + \mathbf{B}_{\text{mask}}), \tag{15}$$

where $\mathbf{W} \in \mathbb{R}^{N \times C}$, $\mathbf{b} \in \mathbb{R}^{L \times L}$, $\mathbf{v} \in \mathbb{R}^{L \times L}$ are learnable parameters that modulate the cross-time dependencies, and $\mathbf{B}_{\text{mask}}$ masks invalid temporal connections.

**Prediction Integration.** The DSTGNN output combines spatial modeling with temporal attention:

$$\hat{\mathbf{y}}_{\text{DSTGNN}} = \mathbf{A}_{\text{coeff}} \odot \mathbf{Z}_{\text{spatial}}. \tag{16}$$

The final prediction integrates hierarchical decomposition with spatio-temporal enhancement:

$$\hat{\mathbf{Y}}_{T+1:T+H} = \hat{\mathbf{Y}}_{\text{NBeats}} + \hat{\mathbf{Y}}_{\text{DSTGNN}}. \tag{17}$$

The unified training strategy ensures that both components learn complementary representations: the white-box blocks focus on structural defined temporal decomposition patterns, while the DSTGNN learns to model residual spatio-temporal dependencies which is more critical to precise prediction of macroeconomic variables. The gradient flows through the entire network, allowing for end-to-end optimization that maximizes the synergy between temporal decomposition and graph-based enhancement.

## 4 EXPERIMENTS

Table 1: Compare EcoXplain's prediction accuracy in GDP forecasts with other baselines

| Country | Metric | ARIMA | N-BEATS | N-BEATSx | Transformer | Informer | Autoformer | FEDformer | Pyraformer | DVGNN | EcoXplain |
|---|---|---|---|---|---|---|---|---|---|---|---|
| China | MAPE | 7.520 | 1.671 | 9.838 | 28.079 | 11.845 | 18.041 | 24.865 | 34.982 | 9.765 | **1.434** |
| | MASE | 0.796 | 0.173 | 1.006 | 2.993 | 1.299 | 1.937 | 2.633 | 3.760 | 1.005 | **0.149** |
| | SMAPE | 7.643 | 1.646 | 9.714 | 33.043 | 12.882 | 20.020 | 29.100 | 43.782 | 9.614 | **1.423** |
| US | MAPE | 0.799 | 0.635 | 52.097 | 19.672 | 12.454 | 24.230 | 4.759 | 16.438 | 1.729 | **0.513** |
| | MASE | 0.459 | 0.379 | 32.356 | 12.394 | 7.929 | 15.240 | 2.971 | 10.527 | 1.039 | **0.304** |
| | SMAPE | 0.805 | 0.636 | 23.997 | 22.048 | 13.466 | 27.888 | 4.934 | 18.407 | 1.746 | **0.514** |
| UK | MAPE | 1.917 | 1.201 | 2.223 | 19.437 | 11.081 | 12.410 | 21.488 | 25.013 | 1.953 | **0.958** |
| | MASE | 1.085 | 0.672 | 1.241 | 11.634 | 6.749 | 7.303 | 12.899 | 14.855 | 1.106 | **0.526** |
| | SMAPE | 1.942 | 1.208 | 2.260 | 21.836 | 11.965 | 13.269 | 24.766 | 28.839 | 1.979 | **0.965** |
| Japan | MAPE | 2.624 | 1.163 | 2.517 | 10.324 | 8.797 | 11.442 | 3.251 | 4.999 | 3.241 | **1.112** |
| | MASE | 0.767 | 0.323 | 0.740 | 3.016 | 2.573 | 3.314 | 0.931 | 1.396 | 0.942 | **0.310** |
| | SMAPE | 2.672 | 1.168 | 2.565 | 11.078 | 9.344 | 12.250 | 3.333 | 4.873 | 3.294 | **1.108** |

### 4.1 EXPERIMENT SETUP

**Dataset Construction.** We have constructed a macroeconomic dataset for the training and testing of EcoTrans. Quarterly time series data on macroeconomic indicators for China, the United States, the United Kingdom, and Japan from 1990 onwards have been collected. It encompasses a comprehensive array of economic sectors, including trade, finance, market dynamics, industry, government affairs, and people's livelihood data for each country. Appendix A.1 presents the types of external data used for GDP forecasting.

**Error Measurement.** To evaluate the performance of the model in practical application scenarios, the macroeconomic data fron the second quarter of 2021 to the first quarter of 2025 (16 consecutive quarters) is designated as the test set. Subsequent evaluations of prediction accuracy in the experiments are conducted based on the test set's error. Three different error evaluation metrics are considered: Symmetric Mean Absolute Percentage Error (SMAPE), Mean Absolute Percentage Error (MAPE), and Mean Absolute Scaled Error (MASE) as described in Appendix A.8.

## 4.2 PREDICTION ACCURACY

In this work, we focus on forecasting the Gross Domestic Product (GDP), one of the most representative macroeconomic indicators to enhance understanding of social economy dynamics. Five Transformer-based models, two white-box models and one spatio-temporal models are employed as baselines for comparison with EcoXplain across the prediction tasks for the four countries, as outlined in Table 1. Across four countries and three error metrics, EcoXplain attains an average MAPE of 1.04%, yielding relative reductions of **88.98%** vs white-box models and **68.07%** vs the spatio-temporal model DVGNN, which delivered the best performance among all black-box baselines. The forecast curves for the GDP data in the test set are depicted in Appendix A.2.

## 4.3 ECONOMIC INTERPRETABILITY

**Interpretable Decomposition.** We added the conventional economics function constraint to substacks to fit a specific inductive paradigm, such as enforcing a Fourier function paradigm to generalize economic cycles.

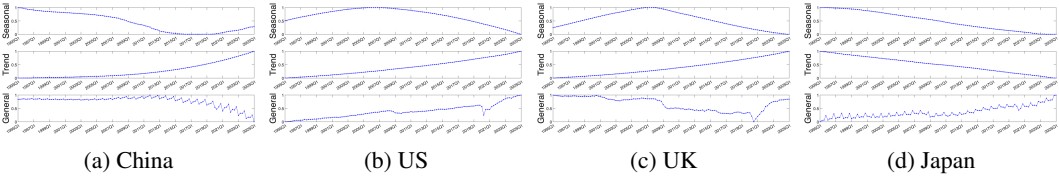

(a) China      (b) US      (c) UK      (d) Japan

Figure 2: The **interpretable decomposition** of EcoXplain provides an elaboration of forecast results consistent with the theoretical framework of economics. The Seasonal, Trend and General curves in each graph represent the normalized forecasting component derived from the S-, T-, and G-Stacks, respectively.

In Figure 2, we illustrate the interpretable decomposition process for the forecast curves in Experiment 4.2. This figure illustrates the diverse economic characteristics of various countries. For instance, the seasonal curves reveal the presence of long-term 'grand cycles' in the GDP of different countries. From 1995 to 2025, each country has experienced approximately half of such a cycle, which aligns with the macroeconomic perspective that a full grand cycle spans about 60 years (Kondratieff, 1925). Moreover, G-Stack in our model effectively captures the cyclical fluctuations in China driven by government regulation, as well as external shocks such as the COVID-19 pandemic, as reflected in the General curves of the US and UK.

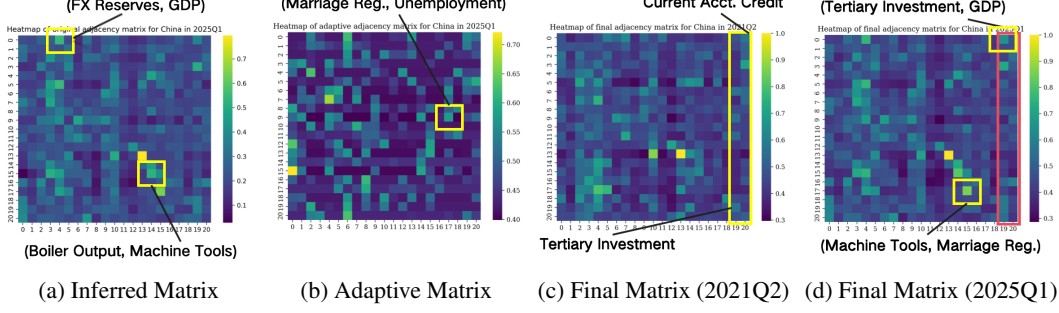

(a) Inferred Matrix      (b) Adaptive Matrix      (c) Final Matrix (2021Q2)      (d) Final Matrix (2025Q1)

Figure 3: Evolution of causal dependency matrices across different stages and time periods in China. Panels (a) and (b) show the 2025Q1 original short-term causalities and adaptive long-term dependencies, while (c) and (d) compare the final combined structures between the pandemic period 2021Q2 and post-recovery 2025Q1.

**Dynamic Causal Graph.** We analyze the estimated causal dependencies between selected Chinese macroeconomic variables across three stages: the *inferred* Granger Causal relations, the *adaptive* global dependencies derived from predictive learning, and the *final* combined structure. The

variable definitions follow Appendix A.5. The strength of each causal link is reported in the range $[0, 1]$.

In the *inference* stage, the most strong short-term linkage is the influence of foreign exchange reserves (4) on GDP (0). This channel has been well documented in the literature: a higher level of reserves enhances financial stability, anchors investor confidence, and provides greater flexibility for monetary policy, which in turn stimulates output growth (Calvo, 2001; Obstfeld et al., 2010). Another significant connection is the co-movement between industrial boiler production (14) and machine tool production (15), which reflects cyclical complementarities in the heavy manufacturing sector.

The *adaptive* stage usually reveals predictive dependencies that are not necessarily grounded in established theory. Interestingly, marriage registrations (17) are found to affect the urban unemployment rate (9), a relationship rarely formalized in economics but plausible in the Chinese context, as marriage often leads to changes in labor force participation—such as women withdrawing from formal employment for family responsibilities—or induces shifts in household demand that reshape employment dynamics.

The *final* combined structure synthesizes short-term mechanisms and adaptive discoveries. In the final causal graph, metal cutting machine tool output (15) is found to feed into marriage registrations (17), suggesting that manufacturing capacity expansion may indirectly shape household formation through income effects, employment prospects, and demographic behavior (Autor et al., 2019). Moreover, investment in the tertiary sector (19) emerges as an important driver of GDP (0), reflecting the growing structural importance of services in China's economy (Chen et al., 2022).

**Evolution Across Time: Evidence from 2021Q2.** During the pandemic period of 2021Q2, the causal matrix exhibits a distinct configuration. Compared with the 2025Q1 structure, where service-sector investment (19) plays a more prominent role, the 2021Q2 causal matrix indicates that current account credit (20) assumed a more central role than in later phases, as reflected by its higher connectivity and stronger influence on other macroeconomic variables. This shift plausibly reflects the contraction of the service sector during the pandemic, which weakened the influence of fixed asset investment in tertiary industries (19) on the broader economy, while external trade and capital flows became more critical for short-term stabilization and economy growth (Beirne et al., 2021).

## 4.4 ROBUSTNESS

In our study, we have conducted a comprehensive analysis of the performance of EcoXplain on various macroeconomic indicators. The results are presented in Table 2, where we have selected challenging Chinese macro indicators as the test subjects. The grey shading of the table items indicates the degree of forecast error, as measured by SMAPE, MAPE, and MASE.

It is worth noting that, despite EcoXplain's remarkable performance, it fails to achieve optimal forecasts for specific indicators, such as CEG[4], where the model struggles to capture short-term fluctuations driven by policy-induced production adjustments, especially when irrelevant domestic external hints are introduced. Moreover, our analysis reveals that EcoXplain exhibits a higher advantage in complex tasks, such as CLB[3] and CGPR[5], where structural dependencies and cross-sectoral linkages are more pronounced.

Table 2: The generalizability of EcoXplain for forecasting various macroeconomic indicator[1].

| Dataset | Metric | Model | | | | | | | | | |
|---|---|---|---|---|---|---|---|---|---|---|---|
| | | ARIMA | N-BEATS | N-BEATSx | Transformer | Informer | Autoformer | FEDformer | Pyraformer | DVGNN | EcoXplain |
| CEV[2] | MAPE | 7.28 | 9.49 | 9.52 | 6.43 | 25.41 | 41.77 | 31.13 | 22.29 | 8.35 | 7.07 |
| | MASE | 0.93 | 1.18 | 1.19 | 0.81 | 3.27 | 5.37 | 3.96 | 2.87 | 0.97 | 0.89 |
| | SMAPE | 7.45 | 9.24 | 9.34 | 6.41 | 29.27 | 55.47 | 38.03 | 25.30 | 8.41 | 7.00 |
| CLB[3] | MAPE | 1.04 | 0.69 | 2.29 | 34.80 | 27.27 | 34.71 | 32.98 | 37.00 | 2.29 | 0.68 |
| | MASE | 0.45 | 0.30 | 0.95 | 15.21 | 11.90 | 14.74 | 14.16 | 15.97 | 1.00 | 0.30 |
| | SMAPE | 1.04 | 0.69 | 2.32 | 43.24 | 32.10 | 44.73 | 40.24 | 45.94 | 2.32 | 0.68 |
| CEG[4] | MAPE | 3.67 | 2.20 | 5.77 | 15.13 | 6.48 | 25.81 | 22.90 | 21.03 | 5.47 | 2.32 |
| | MASE | 0.64 | 0.38 | 1.00 | 2.66 | 1.15 | 4.47 | 3.95 | 3.70 | 0.98 | 0.40 |
| | SMAPE | 3.72 | 2.18 | 5.76 | 16.69 | 6.78 | 30.03 | 26.13 | 24.16 | 5.52 | 2.31 |
| CGPR[5] | MAPE | 12.67 | 13.31 | 15.42 | 14.67 | 18.11 | 31.42 | 22.66 | 28.55 | 17.76 | 12.96 |
| | MASE | 0.72 | 0.73 | 0.86 | 0.88 | 1.11 | 1.79 | 1.33 | 1.72 | 0.95 | 0.71 |
| | SMAPE | 12.68 | 12.34 | 15.20 | 15.96 | 20.41 | 37.94 | 26.42 | 34.22 | 17.60 | 12.32 |

4.5 ABLATION STUDY

To rigorously evaluate the contribution of different components and external information, we design two sets of ablation experiments, all conducted on the global dataset covering China, Japan, the United States, and the United Kingdom:

**Ablation Study on Model Components.** To evaluate the contribution of different components of our framework, we conduct ablation experiments on four variants: the full EcoXplain model, a variant without the white-box decomposition blocks (DSTGNN only), a variant without the DSTGNN module (decomposition only), and a variant without any exogenous information (target data only), corresponding to 0 groups of external variables.

Table 3: Ablation study of model components on forecasting performance

| Model variant | China | Japan | US | UK |
|---|---|---|---|---|
| EcoXplain | **1.43** | **1.11** | **0.61** | **0.96** |
| - DSTGNN only | 9.76 | 3.24 | 1.95 | 1.73 |
| - decomposition only | 1.67 | 1.16 | 0.63 | 1.20 |
| - target data only | 1.94 | 1.16 | 0.82 | 1.03 |

This table shows the performance of different model structures with MAPE%.

The results in Table 3 demonstrate that each component of EcoXplain contributes substantially to forecasting accuracy. Removing the white-box decomposition blocks causes the largest performance deterioration, underscoring the importance of interpretable economic priors for capturing structural regularities in macroeconomic time series. Likewise, excluding either the DSTGNN module or the external data input weakens performance across all four countries, highlighting the necessity of jointly modeling dynamic causal dependencies and exogenous information. Additional ablation experiments varying the number of external variable groups are reported in Appendix A.6.

**Sensitivity to Adaptive Matrix.** We vary the adaptive matrix weight $\alpha \in \{0.0, 0.1, \dots, 1.0\}$ while keeping all other settings fixed, and report the forecasting performance at each $\alpha$. This reveals the model's sensitivity to the balance between prior structural knowledge and adaptive dependencies. On average, the best $\alpha$ in the four countries dataset outperforms the pure inferred matrix method ($\alpha = 0$) by **9.75%** and the pure adaptive matrix method ($\alpha = 1$) by **20.04%**. Details about forecasting accuracy's sensitivity to $\alpha$ are reported in Appendix A.7.

# 5 CONCLUSION

In this paper, we proposed **EcoXplain**, an interpretable framework that integrates economic-based white-box decomposition with a Dynamic Spatio-Temporal Graph Neural Network (DSTGNN) to capture evolving causal dependencies among macroeconomic variables. Through extensive experiments across multiple countries and target variables, we demonstrated that EcoXplain consistently outperforms both classical statistical models and state-of-the-art black-box machine learning approaches, showing particular strength on low-frequency macroeconomic datasets. Beyond improving forecasting accuracy, EcoXplain contributes a methodological foundation for macroeconomic analysis by jointly leveraging short-term causal structures inferred within sliding windows and prediction-driven adaptive relationships that uncover previously hidden interdependencies. This dual perspective not only enhances predictive performance but also provides practical value for policymakers, offering insights into how interventions and shocks may propagate across the economy.

---

[1] The dark grey items present the method with the highest predictive accuracy, while the light grey items denote the second-best predictive method. Note that the table rounds all numbers to two decimal places; the ranking of best and second-best methods was determined using more precise underlying values.

[2] **CEV**: China's Export Value (billions of USD, quarterly).

[3] **CLB**: Total Loan Balance of China's Financial Institutions (trillions of RMB, quarterly).

[4] **CEG**: China's Electricity Generation (terawatt-hours, monthly aggregated to quarterly).

[5] **CGPR**: China's General Public Budget Revenue (billions of RMB, quarterly).

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

## A  APPENDIX

### A.1  DATASET DESCRIPTION

In our experiment dataset, quarterly time series data on macroeconomic indicators for China, the United States, the United Kingdom and Japan from 1990 onwards have been collected. It encompasses a comprehensive array of economic sectors, including trade, finance, market, industry, government, society, etc. for each country. Table 4 presents details about the typical types of external data used for GDP forecasting.

Table 4: Exogenous macroeconomic indicators data set

| Category | Exogenous Data | Country | Unit |
|---|---|---|---|
| Trade | Exchange Rate (Average USD to CNY) | China | CNY/USD |
| | Trade (Exports / Imports / Balance) | China, Japan, UK, USA | 100 Million USD |
| Finance | Consumer Price Index (CPI) | China, USA, Japan | index |
| | CPI by category (YoY / MoM) | China, USA | % |
| | Producer Price Index (PPI, all goods) | China, Japan, USA | index |
| | Contribution to YoY Growth of Real GDP | Japan | % |
| | M2 Money Supply | Japan, UK, USA | Million USD |
| | Fixed Asset Investment Completion Amount | China | 100 Million CNY |
| Market | Total Retail Sales of Consumer Goods | China, UK, USA | 100 Million Local Currency |
| | Commercial Sales Amount | Japan | Billion JPY |
| | Benchmark Yield of 10-Year Treasury Bonds | UK, Japan | % |
| | Total Deposits Balance of Financial Institutions | China | 100 Million CNY |
| | Total Loans Balance of Financial Institutions | China | 100 Million CNY |
| | Total Consumer Credit | USA | Million USD |
| | Consumer Confidence Index | China, Japan, UK | index |
| | Housing Loans (new and outstanding) | Japan | 100 Million JPY |
| | OECD Real House Price Index | Japan | index |
| Industry | Industrial Production Index | UK, USA | index |
| | Industrial Production Index (YoY) | Japan | % |
| | Industrial Added Value (YoY) | China | % |
| | Manufacturing Inventories | Japan | 100 Million JPY |
| | Producer Shipment Index | Japan | index |
| | Capacity Utilization Rate | China | % |
| | Electricity Generation (YoY) | China, USA | % / index |
| | Steel Production | China, Japan, USA | 10 kt |
| | Automobile Production | China, Japan, UK, USA | 10 Thousand Units |
| | Household Refrigerator Production | China | 10 Thousand Units |
| | Metal-Cutting Machine Tool Production / Orders | China, Japan | 10 Thousand Units / Million JPY |
| Government | Fiscal Balance | China | 100 Million CNY |
| | Public Finance Revenue | China | 100 Million CNY |
| | Public Finance Expenditure | China | 100 Million CNY |
| | Tax Revenue | China, UK, USA | 100 Million CNY / Million GBP / Million USD |
| | Government Debt and Liabilities | China, Japan, UK | 100 Million Local Currency / Million GBP |
| Society | Completed Housing Area (YoY) | China | % |
| | Housing Area under Construction (YoY) | China | % |
| | Newly Started Housing Area (YoY) | China | % |
| | Commodity Housing Sales Area (YoY) | China | % |
| | Employment (Aged 16 and above) | UK, USA | Thousand People |
| | New Non-Agricultural Employment (by sector) | USA | Thousand People |
| | Newly employed persons in urban areas | China, UK | Thousand People |
| | Unemployment Rate (total, male, female) | Japan | % |
| | Unemployed Population | UK | Thousand People |
| | Number of People Applying for Unemployment Benefits | UK | Thousand People |
| | Average Weekly Working Hours of Manufacturing Industry | USA | Hour |
| | Real Personal Disposable Income Index | Japan | index |
| Comprehensive | Current Account Balance | Japan | 100 Million JPY |
| | OECD Composite Leading Indicators | Japan | index |
| | Patents (Granted / In Force) | China, Japan | Units |
| | Financial Market Indicators (SCI, TSE, NASDAQ, Dow Jones) | Japan, USA, China | index |

## A.2 PREDICTION ACCURACY

The GDP prediction curves among four countries are shown in Figure 4. As illustrated, EcoXplain provides more accurate estimates of economic trends, cycles and disruptions than other baselines, particularly during the COVID-19 pandemic when fluctuations were most pronounced.

## A.3 CAUSAL INFERENCE METHODOLOGY

In this section we provide details of the two-stage causal inference procedure adopted in our model, which basically follows the framework of CUTS (Cheng et al., 2023) to handle time-varying eco-

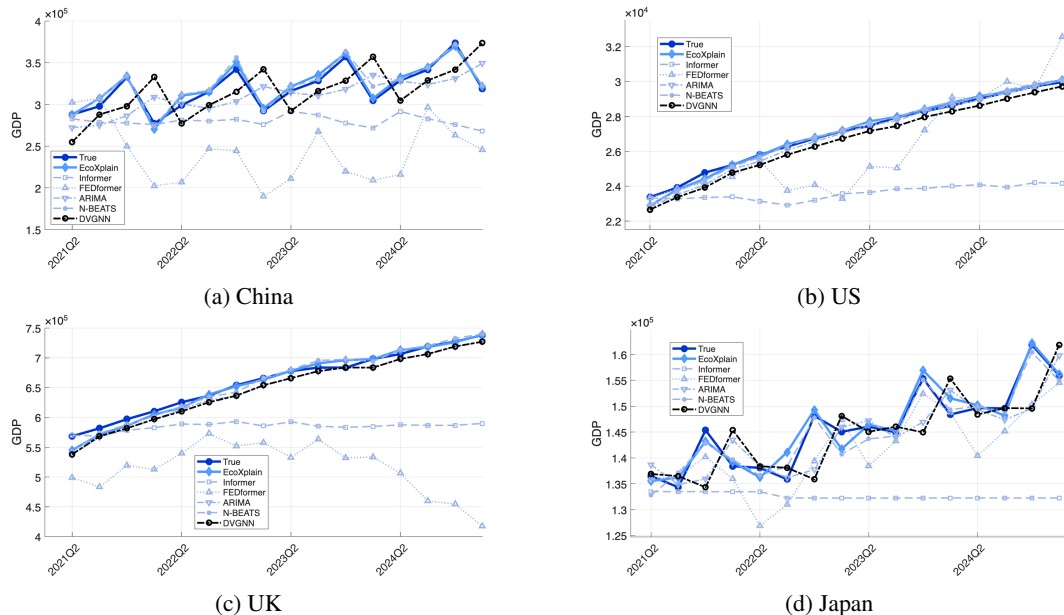

Figure 4: Comparison of GDP forecast curves in the test set (2021Q2–2025Q1) for four countries using various macroeconomic forecasting methods.

nomic relationships. Our approach constructs dynamic causal graphs that capture evolving Granger-type dependencies between macroeconomic variables across different time periods.

### A.3.1 DYNAMIC CAUSAL GRAPH LEARNING ALGORITHM

The dynamic causal graph learning process consists of three main stages adapted for time-varying economic systems:

**Stage 1: Latent Data Prediction.** For each time window $t$, we employ a Delayed Supervision Graph Neural Network (DSGNN) to predict missing values, by sampling the causal graph with Bernoulli distribution (Lippe et al., 2022):

$$\hat{x}_{t,i}^{(t)} = f_{\phi_i}(X^{(t)} \odot S^{(t)}) = f_{\phi_i}(x_{t-\tau:t-1,1}^{(t)} \odot s_{1:\tau,1i}^{(t)}, \ldots, x_{t-\tau:t-1,N}^{(t)} \odot s_{1:\tau,Ni}^{(t)}) \tag{18}$$

where $S^{(t)} = \{S_\tau^{(t)}\}_{\tau=1}^{\tau_{max}}$ and $s_{\tau,ij}^{(t)} \sim \text{Ber}(1 - m_{\tau,ij}^{(t)})$. The Hadamard product $\odot$ masks the causal connections according to the sampled graph structure.

**Stage 2: Causal Graph Discovery** The causal graph parameters are optimized by minimizing:

$$\mathcal{L}_{\text{graph}}^{(t)}(\tilde{X}^{(t)}, \hat{X}^{(t)}, O^{(t)}, \theta^{(t)}) = \mathcal{L}_{\text{pred}}^{(t)}(\tilde{X}^{(t)}, \hat{X}^{(t)}, O^{(t)}) + \lambda||\sigma(\theta^{(t)})||_1 \tag{19}$$

where the prediction loss is defined as:

$$\mathcal{L}_{\text{pred}}^{(t)}(\tilde{X}^{(t)}, \hat{X}^{(t)}, O^{(t)}) = \sum_{i=1}^{N} \frac{\langle \mathcal{L}_2(\hat{x}_{1:L,i}^{(t)}, \tilde{x}_{1:L,i}^{(t)}), o_{1:L,i}^{(t)} \rangle}{\frac{1}{L}\langle o_{1:L,i}^{(t)}, o_{1:L,i}^{(t)} \rangle} \tag{20}$$

The $L_1$ regularization term $\lambda||\sigma(\theta^{(t)})||_1$ enforces sparsity in the causal connections, preventing overfitting and ensuring that only significant causal relationships are retained. The $\mathcal{L}_2$ represents the MSE Loss in the prediction stage.

**Stage 3: Iterative Data Imputation** Missing values are iteratively updated using:

---

**Algorithm 1:** Dynamic Causal Graph Construction

---

**Input:** Time series data $X$, window size $w$, maximum lag $\tau_{\max}$

**Output:** Dynamic causal graphs $\{\mathbf{A}_{\text{dynamic}}^{(t)}\}_{t=w}^{T}$

**1 for** $t = w$ **to** $T$ **do**

2     Extract window data: $X^{(t)} = X[t - w + 1 : t, :]$;

3     Initialize causal parameters: $\theta_{\tau,ij}^{(t)}$ for all $i, j, \tau$;

**4**     **for** $epoch = 1$ **to** $n_1 + n_2 + n_3$ **do**

**5**        **if** $epoch \leq n_1$ **then**

6           Train without imputation;

**7**        **else if** $epoch \leq n_1 + n_2$ **then**

8           Train with imputation but no supervision from imputed data;

**9**        **else**

10          Fine-tune with supervision from all data (including imputed);

11     Update $\theta^{(t)}$ and neural network parameters;

12     Compute causal probabilities: $m_{\tau,ij}^{(t)} = \sigma(\theta_{\tau,ij}^{(t)})$;

13     Construct adjacency matrix: $\tilde{a}_{i,j}^{(t)} = \max_{\tau \in \{1,\ldots,\tau_{\max}\}} m_{\tau,ij}^{(t)}$;

---

$$\tilde{x}_{t,i}^{(t,m+1)} = \begin{cases} (1 - \alpha)\tilde{x}_{t,i}^{(t,m)} + \alpha\hat{x}_{t,i}^{(t,m)} & \text{if } o_{t,i}^{(t)} = 0 \text{ and } m \geq n_1 \\ \tilde{x}_{t,i}^{(t,0)} & \text{if } o_{t,i}^{(t)} = 1 \text{ or } m < n_1 \end{cases} \tag{21}$$

where $m$ indexes the iteration steps, $\alpha$ is a smoothing parameter to prevent abrupt changes, and $n_1$ is the number of initial epochs without data imputation.

### A.3.2 CAUSAL PROBABILITY MODELING

The causal probability $m_{\tau,ij}^{(t)}$ represents the likelihood that time-series $i$ at lag $\tau$ causally influences time-series $j$ at time $t$. Following the methodology from the original DSGNN framework, we model this probability using the sigmoid function:

$$m_{\tau,ij}^{(t)} = \sigma(\theta_{\tau,ij}^{(t)}) \tag{22}$$

where $\sigma(\cdot)$ denotes the sigmoid function and $\theta_{\tau,ij}^{(t)}$ is the learned parameter that captures the strength of the causal relationship. The sigmoid function ensures that $m_{\tau,ij}^{(t)} \in [0, 1]$, providing a probabilistic interpretation of causal strength.

### A.3.3 SLIDING WINDOW IMPLEMENTATION

Algorithm 1 summarizes our sliding window implementation for constructing dynamic causal graphs across the entire time series. The algorithm processes the time series sequentially, learning causal relationships within each sliding window, which enables our model to capture short-term fluctuations in macroeconomic causal relationships for each period.

### A.4 PSEUDOCODE FOR ECOXPLAIN

To enhance clarity and reproducibility, we provide the pseudocode of our proposed framework in this section. The algorithms are presented in a modular form: Algorithm 2 outlines the overall EcoXplain training loop, where backcast–forecast decomposition is iteratively refined through block-wise updates. Algorithm 3 details how the inferred dynamic graphs from Algorithm 1 are combined with adaptive adjacency matrices to form the spatio-temporal dependencies exploited by DSTGNN. Within DSTGNN, Chebyshev graph convolution captures higher-order spatial interactions, while dual-branch temporal attention adaptively weights heterogeneous time dependencies. Together,

---

**Algorithm 2:** EcoXplain Main Structure

---

**Input:** Dataset $X$ (macroeconomic data), Dataset $EX$ (exogenous data), backcast length $b$, forecast length $f$, Stack/Block configs, adaptive weight $\alpha$

1 **while** *current_epoch < epochs* **do**
2    $index \leftarrow$ random window indices;
3    $y \leftarrow X[index(-1) + 1 : index(-1) + f]$;
4    $EX_{in} \leftarrow EX[index(0) : index(-1)]$;
5    $X_{in} \leftarrow X[index(0) : index(-1)]$;
6    $res \leftarrow X_{in}$;
7    **while** $i < \#blocks$ **do**
8       **if** $Stack == S$ **then**
9          $(fore, back) \leftarrow Block_S(res)$ ;        // seasonality constraint
10       **else if** $Stack == T$ **then**
11          $(fore, back) \leftarrow Block_T(res)$ ;         // trend constraint
12       **else if** $Stack == G$ **then**
13          $(fore, back) \leftarrow Block_G(res)$ ;         // generic constraint
14       **else**
15          $fore \leftarrow$ **DSTGNN**$(res, EX_{in}, \alpha)$ ;    // call Algorithm 3
16          $back \leftarrow None$;
17       $res \leftarrow res - back$;
18       $y' \leftarrow X_{in}[-1] + fore$;
19       $i + +$;
20    Compute loss $\mathcal{L}(y, y')$ (MAPE) and backpropagate;
21    $current\_epoch + +$;

---

**Algorithm 3:** DSTGNN Prediction with Dynamic Causal Graph Construction

---

**Input:** Input tensor $\mathbf{R}_t \in \mathbb{R}^{B \times C \times N \times L}$, exogenous data $EX_{in}$, adaptive weight $\alpha$, Chebyshev order $K$, temporal kernel $K_t$, prediction step $T$

**Output:** Forecast $\hat{\mathbf{y}}_{\text{DSTGNN}} \in \mathbb{R}^{B \times N \times f}$, final adjacency $\mathbf{A}^{(t \to T)}$, adaptive adjacency $\mathbf{A}_{\text{adapt}}^{(T)}$

1 Initialize learnable embeddings $\mathbf{E}_1^{(T)}, \mathbf{E}_2^{(T)} \in \mathbb{R}^{N \times d}$;
2 $\mathbf{A}_{\text{adapt}}^{(T)} \leftarrow \text{softmax}\big( \text{ReLU}(\mathbf{E}_1^{(T)} \mathbf{E}_2^{(T)\top}) \big)$ ;      // adaptive: $N \times N$
3 $\mathbf{A}^{(t \to T)} \leftarrow \alpha\, \mathbf{A}_{\text{adapt}}^{(T)} + (1-\alpha)\, \mathbf{A}_{\text{dynamic}}^{(t)}$ ;   // call Algorithm 1, output $\mathbf{A}_{\text{dynamic}}^{(t)}$
4 $\mathbf{A}^{(t \to T)} \leftarrow \mathbf{A}^{(t \to T)} / (\mathbf{1} \mathbf{A}^{(t \to T)} + \varepsilon)$ ;      // row-normalize
5 $\mathbf{L}_0 \leftarrow \mathbf{I}_N$,   $\mathbf{L}_1 \leftarrow \mathbf{A}^{(t \to T)}$;
6 **for** $k = 2$ **to** $K - 1$ **do**
7    $\mathbf{L}_k \leftarrow 2\, \mathbf{A}^{(t \to T)} \mathbf{L}_{k-1} - \mathbf{L}_{k-2}$;
8 $\mathbf{H}_{\text{graph}} \leftarrow \sum_{k=0}^{K-1} \mathbf{L}_k \cdot \mathbf{R}_t \cdot \mathbf{W}_k$ ;      // basis-weighted aggregation
9 $\mathbf{Z}_{\text{spatial}} \leftarrow \text{Conv2D}_{\text{temp}}(\mathbf{H}_{\text{graph}}; \mathbf{W}_{\text{temp}})$
10 **for** $j \in \{1, 2\}$ **do**
11    $\mathbf{R}_t^{\text{perm}\, j} \leftarrow \text{permute}_j(\mathbf{R}_t)$;
12    $\mathbf{F}_j \leftarrow \text{Conv2D}_j(\mathbf{R}_t^{\text{perm}\, j}; \mathbf{W}_j)$;
13 $\mathbf{S} \leftarrow \sigma\big((\mathbf{F}_1 \mathbf{W}_{\text{att}})\, \mathbf{F}_2^\top + \mathbf{b}\big)$ ;  // $\mathbf{S} \in \mathbb{R}^{B \times L \times L}$, $\mathbf{W}_{\text{att}} \in \mathbb{R}^{N \times N}$, $\mathbf{b} \in \mathbb{R}^{L \times L}$
14 $\mathbf{A}_{\text{coeff}} \leftarrow \text{softmax}(\mathbf{v} \odot \mathbf{S} + \mathbf{B}_{\text{mask}})$ ;      // $\mathbf{v} \in \mathbb{R}^{L \times L}$
15 $\tilde{\mathbf{Z}} \leftarrow \mathbf{A}_{\text{coeff}} \odot \mathbf{Z}_{\text{spatial}}$ ;      // elementwise temporal weighting
16 $\hat{\mathbf{y}}_{\text{DSTGNN}} \leftarrow \text{ConvHead}(\tilde{\mathbf{Z}})$
17 **return** $\hat{\mathbf{y}}_{\text{DSTGNN}}$, $\mathbf{A}^{(t \to T)}$, $\mathbf{A}_{\text{adapt}}^{(T)}$

---

these components constitute the core workflow of EcoXplain, providing a transparent view of how causal structure learning, adaptive graph refinement, and temporal modeling are integrated into the final forecasting process.

## A.5 EXTERNAL VARIABLE USED IN ANALYSIS

In order to capture rich macroeconomic dynamics, beyond the external data in experiment dataset listed above, EcoXplain incorporates a wide range of meticulously-selected exogenous variables. Table 5 lists the variables used in the analysis, covering key macro-financial indicators such as

monetary policy, trade, production, and labor market conditions. These external variables provide complementary information beyond autoregressive signals, helping the model identify structural drivers of macroeconomic fluctuations.

Table 5: List of External Variables

| ID | Variable |
|---|---|
| 0 | GDP (per capita disposable income used as proxy) |
| 1 | Per capita disposable income of residents (cumulative real YoY) |
| 2 | Coal output (current month) |
| 3 | Reserve money (base money) |
| 4 | Official reserve assets: foreign exchange reserves |
| 5 | Output of household refrigerators (current month) |
| 6 | New RMB loans: national small and medium-sized banks (cumulative) |
| 7 | Shanghai Stock Exchange: total market capitalization |
| 8 | Patent applications granted: invention patents (domestic, current month) |
| 9 | Urban surveyed unemployment rate |
| 10 | Leverage ratio of the real economy sector |
| 11 | Integrated circuit output (current month) |
| 12 | CPI: food (YoY, current month) |
| 13 | CPI: services (YoY, current month) |
| 14 | Industrial boiler output (current month) |
| 15 | Metal cutting machine tool output (current month) |
| 16 | Newly employed persons in urban areas (cumulative) |
| 17 | Marriage registrations: number of marriages (cumulative) |
| 18 | Central government debt balance (total) |
| 19 | Fixed asset investment completed: tertiary industry (cumulative) |
| 20 | Current account credit (quarterly) |

## A.6 SENSITIVITY TO EXTERNAL VARIABLES

Beyond the structural ablations reported in the main paper (Table 3), we further analyze the sensitivity of EcoXplain to the number of external variable groups included. With $\alpha$ fixed at 0.5, we randomly select 10, 20, and 30 groups of external variables. Each configuration is repeated five times with different random draws, and we report the average forecasting performance (MAPE).

Table 6: Effect of the number of external variable groups on forecasting performance. Measured in MAPE(%). Each setting is averaged over 5 runs with different random draws.

| External groups | China | Japan | US | UK |
|---|---|---|---|---|
| 0 (no external) | 1.94 | 1.16 | 0.82 | 1.03 |
| 10 groups (5 runs) | 1.63 | 1.14 | 0.69 | 1.02 |
| 20 groups (5 runs) | 1.52 | 1.11 | 0.68 | **1.01** |
| 30 groups (5 runs) | **1.46** | **1.11** | **0.64** | 1.04 |

As shown in Table 6, incorporating exogenous variables generally improves predictive accuracy. However, the performance gain does not increase monotonically with the number of groups. With a fixed sample size, introducing more external variables makes it harder to reliably capture interdependencies, and the adaptive matrix is further constrained by its dimensionality $d$. This can reduce performance when the number of variables becomes too large, suggesting the need for a balanced choice of external information.

## A.7 SENSITIVITY TO ADAPTIVE MATRIX

Figure 5 reports the forecasting accuracy measured by MAPE as the adaptive weight $\alpha$ varies from 0.0 to 1.0. The results highlight that the optimal balance between dynamic causal graphs and adaptive adjacency differs across countries. The best $\alpha$ values are 0.4 for China, 0.5 for Japan, 0.8 for the

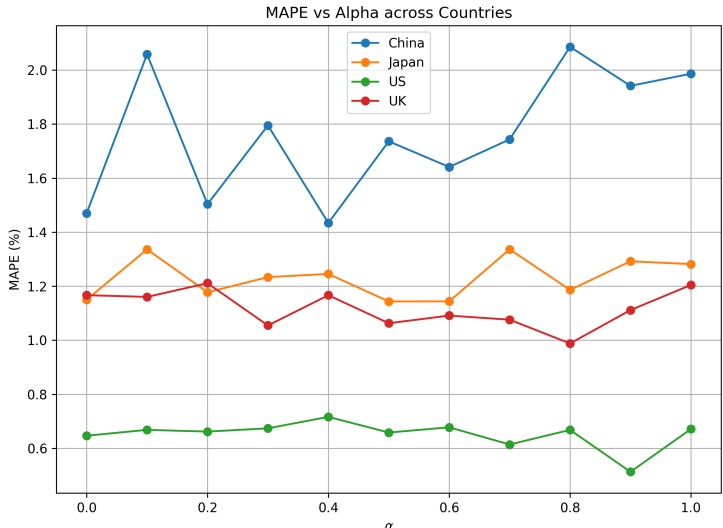

Figure 5: Sensitivity of forecasting performance to the adaptive weight $\alpha$ across four countries. Each curve shows MAPE as $\alpha$ varies from 0 (pure dynamic causal graph) to 1 (pure adaptive adjacency).

UK, and $0.9$ for the US. Compared with the extreme cases ($\alpha = 0$ using only dynamic graphs and $\alpha = 1$ using only adaptive adjacency), the optimal $\alpha$ consistently improves forecasting accuracy: for instance, gains of $27.8\%$ in China and $23.6\%$ in the US are observed relative to the fully adaptive variant. These results confirm that EcoXplain benefits most from a hybrid structure, where prior causal knowledge and adaptive dependencies are jointly exploited.

## A.8 ERROR MEASUREMENT

The prediction error measurement metrics are defined as follows:

$$\text{SMAPE} = \frac{100\%}{n} \sum_{t=1}^{n} \frac{|y_t - \hat{y}_t|}{\frac{|y_t| + |\hat{y}_t|}{2}}, \tag{23}$$

$$\text{MAPE} = \frac{100\%}{n} \sum_{t=1}^{n} \left| \frac{y_t - \hat{y}_t}{y_t} \right|, \tag{24}$$

$$\text{MASE} = \frac{\frac{1}{n} \sum_{t=1}^{n} |y_t - \hat{y}_t|}{\frac{1}{n-1} \sum_{t=2}^{n} |y_t - y_{t-1}|}, \tag{25}$$

where $y_t$ and $\hat{y}_t$ denote the ground-truth and predicted values at time $t$, respectively, and $n$ represents the number of test samples.

## A.9 USE OF LARGE LANGUAGE MODELS

Large language models (LLMs) were employed strictly as auxiliary tools in preparing this paper. Their use was limited to:

- **Editorial Support:** Assisting with grammatical corrections, wording refinements, and stylistic improvements to enhance clarity.
- **Presentation Support:** Providing suggestions for table layouts and figure captions to improve readability and consistency.

LLMs did not contribute to research design, analysis, or the development of core technical ideas. The authors bear full responsibility for the content of the paper, and LLMs are not considered contributors or co-authors.

