# OpenReview forum: "EcoXplain: An Interpretable Causal-Augmented Framework for Macroeconomic Forecasting"
_ICLR.cc/2026/Conference — ICLR 2026 Conference Withdrawn Submission_

### Official Review · Reviewer_B9JP · 2025-11-01

**Soundness:** 3
**Presentation:** 3
**Contribution:** 2
**Rating:** 2
**Confidence:** 3

**Summary:**

This paper proposes EcoXplain, an interpretable framework designed for explaining variations in low-frequency macroeconomic data with a Dynamic Spatio-Temporal Graph Neural Network (DSTGNN), which integrates short-term inferred dynamic causal relationships with prediction-driven adaptive adjacency matrices that capture evolving relationships between macroeconomic variables.

**Strengths:**

This paper addresses an important problem of building a spatio temporal neural network model to explain variations in macroeconomic data. The problem has high significance for policy makers and economists.

**Weaknesses:**

The underlying algorithmic contributions are well known in the literature which may limit the algorithmic novelty of the work.

Many of the variables modeled are changing only once per quarter and even if you split it across several years spanning a decade, the number of observation points are so small that significantly weakens the "causality" or "causal aware" arguments of the paper. Many of these models are fitting heavy weight models for a much smaller number of data points and variables.

The paper may have missed out on several non-linear time-series papers in the ML + Economics literature and many of these paper have proposed much fancier models with a significant broader array of signals.

The input vector space is actually quite small. Its almost looking like a simple set of strongly correlated economic variables. Yes a non-linear model can provide better MAPE and MAE values but it could also be a classic overfitting issue with limited data.

Ideally, for an analysis of this scale, the paper also needs to demonstrate auxiliary evidence of correctness of causal signals extracted or claimed. For example, prior work have also looked into news as an auxiliary source.

The MAPE, MAE variability across the algorithms are very high which raises questions on experimental setup. Some of these techniques should not yield such high error even for relatively simplistic generalized linear models using the variables.

**Questions:**

The paper may have missed out on several non-linear time-series papers in the ML + Economics literature and many of these paper have proposed much fancier models with a significant broader array of signals. I am happy to see a good mix of ML and Econ papers on this topic but the related work is missing a fairly large body of work in this intersection. The paper may have missed out on several non-linear time-series papers in the ML + Economics literature and many of these paper have proposed much fancier models with a significant broader array of signals.

Here are some authors to check out:
David Lazer (Northeastern)
Samuel Fraiberger (World Bank DIME)
Ananth Balashankar (Google Deepmind)
Joshua Blumenstock (Berkeley)

Please comment on the volume of "predicted" data in training vs test as well as contrast them with how the test split was done to answer the concern below:
Many of the variables modeled are changing only once per quarter and even if you split it across several years spanning a decade, the number of observation points are so small that significantly weakens the "causality" or "causal aware" arguments of the paper. Many of these models are fitting heavy weight models for a much smaller number of data points and variables. I would like to get a deeper understanding of why simpler models arent enough for this task. The underlying matrix of all variables with a time-lagged PCA already gives a lot of information of time-lagged correlated signals.


Please comment on your analysis structure to answer this query:
The input vector space is actually quite small. Its almost looking like a simple set of strongly correlated economic variables. Yes a non-linear model can provide better MAPE and MAE values but it could also be a classic overfitting issue with limited data.

Please comment on whether external data can be helpful for validation:
Ideally, for an analysis of this scale, the paper also needs to demonstrate auxiliary evidence of correctness of causal signals extracted or claimed. For example, prior work have also looked into news as an auxiliary source.

---

### Official Review · Reviewer_oi4E · 2025-11-01

**Soundness:** 1
**Presentation:** 1
**Contribution:** 1
**Rating:** 0
**Confidence:** 5

**Summary:**

The manuscript presents a manuscript for time series analyses.
The idea combines interesting and recent AI approaches (e.g., causal inference, GNN) to improve the performance of time series.
Experiments were conducted on real-world data, and using different models.

**Strengths:**

The manuscript is well-written, and there is a notable effort to utilize recent and significant AI tools to address an important issue.The experimental setup presents a very nice branch of methods as benchmarks, and an attempt is made to perform an ablation study.The adoption of mathematical formulation to define some approaches was also important to show deep knowledge of the methods.

**Weaknesses:**

Although the manuscript has great investigative potential, there is a substantial requirement to consider it ready for publication. The main limitations of this work are:
- The main contribution of the manuscript, as read in the title, abstract, and introduction, is the possibility of interpretation. However, the interpretability was not properly explored. Considering only trend and seasonality is not enough to reach the expected goals. I recommend a deep reading of the manuscript about interpretability to better describe the proposal and evaluate it in-depth.
- Most of the effort in the result section was dedicated to superficially mentioning that the proposal achieved reduced prediction errors. It is not totally aligned with the abstract and introduction.
- There is a great effort to show the proposal in macroeconomic forecasting. It is unclear what kind of information this application needs to make the proposal especially necessary.
- Different methods were considered without a clear justification, even considering some methods that are not totally appropriate to this goal.

**Questions:**

- Some equations were used to describe some well-known concepts and approaches, but there is no clear connection between them.
- The introduction is very superficial. This could be further explored through a technical approach to the contribution.
- No actual interpretability was explored in the manuscript. Only analyzing seasonal and trend components is not enough. Why did the authors not select some time series and explore the interpretability provided by the approach?
- It is not clear whether all evaluated methods were properly adjusted to reach the best performance.
- Some equations must be revisited for a better presentation. For example, it is not clear the definition of some terms, such as "v" in Equation 2, and the same with theta in Equation 4, etc.
- Figure 1 is very important, but why did the author discuss in depth each part of their contribution? It is not an easy figure to read.
- The authors emphasize that the contribution was designed for macroeconomic forecasting. Why is it so suitable for this specific application? What kind of information does this application show that is more favorable to the proposal?
- Figure 3 is not simple to understand.

---

### Official Review · Reviewer_Cgcw · 2025-11-01

**Soundness:** 2
**Presentation:** 2
**Contribution:** 3
**Rating:** 4
**Confidence:** 3

**Summary:**

This paper introduces a novel framework called EcoXplain, which combines interpretable white-box decomposition and dynamic spatiotemporal graph neural networks (DSTGNN) to enhance macroeconomic forecasting capabilities. In predicting GDP for the four major economies (China, the US, the UK, and Japan), the model outperforms traditional white-box models and state-of-the-art black-box models, significantly improving the accuracy of prediction errors such as MAPE and SMAPE. EcoXplain also provides economic interpretability by revealing causal relationships between macroeconomic variables, offering valuable insights for policymakers. Ablation experiments highlight the importance of each component of the model, confirming that both white-box decomposition and dynamic causal graph fitting contribute to improving the model's predictive power. However, this paper lacks a detailed explanation of how the dataset was constructed, does not clarify whether the dataset covers the full range of macroeconomic conditions across different economies, and addresses other issues requiring improvement.

**Strengths:**

1. The paper introduces EcoXplain, an innovative hybrid framework that combines interpretable white-box decomposition with a Dynamic Spatio-Temporal Graph Neural Network (DSTGNN), offering a novel approach to capturing evolving causal dependencies in macroeconomic forecasting.

2. The methodology is rigorously developed, with a clear focus on improving both the accuracy and interpretability of macroeconomic forecasts, demonstrated through extensive experiments across multiple countries.

3. The paper is well-structured, with a detailed explanation of the EcoXplain framework, clear definitions of each model component, and comprehensive experimental results that are easy to follow.

**Weaknesses:**

1. While the authors provide diverse datasets from countries such as China, the United States, the United Kingdom, and Japan, the paper lacks in-depth explanation of how these countries were chosen, and fails to demonstrate whether the datasets truly represent the full picture of the global macroeconomic situation.

2. Although EcoXplain outperforms benchmark methods such as N-BEATS and Transformer, the paper does not adequately explore the trade-off between model complexity and predictive accuracy. Complex models like EcoXplain may offer higher accuracy but may also suffer from overfitting, high computational costs, or difficulty in implementing them in real-world environments.

3. While the paper demonstrates some economic interpretability through the decomposition of predictive components, it does not delve into how the causal relationships revealed by EcoXplain inform economic policy. The analysis of causal graphs and dependencies is also rather brief and requires further exploration. Furthermore, the paper does not demonstrate the robustness and stability of these causal relationships under different economic conditions.

4. This paper relies on Granger causality to identify relationships between economic variables, but Granger causality only assesses statistical correlation, not true causal relationships. The authors did not adequately explore the potential limitations of using Granger causality to capture nonlinear, time-varying, or complex causal dependencies.

5. This paper does not evaluate the model's performance under conditions of high data noise or incompleteness. Macroeconomic data is typically complex and susceptible to sudden external shocks, which may affect the model's stability and generalization ability.

**Questions:**

Refer to Weaknesses.

---

### Official Review · Reviewer_m3qC · 2025-11-01

**Soundness:** 3
**Presentation:** 3
**Contribution:** 3
**Rating:** 6
**Confidence:** 4

**Summary:**

This paper proposes EcoXplain, an interpretable causal-augmented spatio-temporal forecasting framework for macroeconomic prediction.
The model integrates a white-box decomposition module (inspired by N-BEATS) with a Dynamic Spatio-Temporal Graph Neural Network (DSTGNN) that fuses short-term Granger causal graphs and adaptive adjacency matrices learned during prediction. Experiments on datasets from China, Japan, the US, and the UK show substantial performance gains (up to 88.98 % vs. white-box and 68.07 % vs. black-box baselines). Beyond accuracy, EcoXplain provides interpretable trend/seasonality decomposition and evolving causal matrices that reveal meaningful economic relationships (e.g., investment–GDP linkages, pandemic-period causal shifts). Overall, the work bridges causal interpretability and deep forecasting in macroeconomics, with both methodological and practical value.

**Strengths:**

Innovation: Creative combination of N-BEATS-style white-box blocks with DSTGNN causal modeling.

Interpretability: Clear trend/seasonality decomposition and causal-path visualization support economic reasoning.

Empirical strength: Large performance gains across multiple economies and baselines.

Relevance: Addresses an important challenge—making deep macroeconomic forecasting both accurate and interpretable.

Completeness: Includes ablations, sensitivity studies, and thorough appendix detailing data and pseudocode.

**Weaknesses:**

Causal rigor: The causal discovery stage (based on Granger/CUTS) may not ensure structural causality; clarify assumptions and limitations.

Scalability: Runtime and memory cost of dynamic graph updates are not discussed.

Limited target diversity: Main experiments focus on GDP; adding CPI/unemployment would improve generality.

Dense exposition: Mathematical details could be streamlined with more intuition or a high-level pipeline diagram.

**Questions:**

How robust is the causal inference stage to noisy or missing macro data, especially given quarterly frequency?

What prevents the adaptive adjacency from overfitting short-term noise rather than genuine causal shifts?

Can the authors quantify computational cost versus Transformer baselines?

Would EcoXplain still outperform if trained on smaller single-country datasets (data-scarce regimes)?

Could future work extend EcoXplain to counterfactual policy simulation using the learned causal graph?

---

### Official Review · Reviewer_R8xN · 2025-11-04

**Soundness:** 2
**Presentation:** 3
**Contribution:** 3
**Rating:** 2
**Confidence:** 4

**Summary:**

The authors propose an interpretable dynamic causal-enhanced spatiotemporal GNN architecture designed for low-frequency macroeconomic data. This approach addresses the limitations of traditional models that struggle to adapt to rapidly evolving causal dynamics, as well as those of machine learning methods that often lack interpretability.

**Strengths:**

S1. The authors design DSTGNN to achieve accurate forecasting while incorporating interpretable causal mechanisms, which is crucial for economic policy analysis as well as time-series forecasting across various domains.

S2. The authors provide reproducible code and datasets.

**Weaknesses:**

W1. It remains unclear whether DSTGNN is intended to address the challenge of fast-moving dynamics or the characteristics of low-frequency data. The abstract mentions that traditional models struggle to adapt to fast-moving causal dynamics, while DSTGNN is specifically designed for low-frequency macroeconomic data.

W2. The authors should provide a clear definition of the term “black box”, as graph-based and attention-based models can offer a certain degree of interpretability, despite being categorized as machine learning methods.

W3. There are minor typographical errors, such as “tne” in Line 236. In addition, the author should specify the unit of MAPE (i.e., %) in Table 1.

W4. The proposed method appears incremental, and its performance improvement relies heavily on the inclusion of external data. The author should include TimeXer [1] as an additional benchmark model.

W5. The results for N-BEATSx and N-BEATS appear inconsistent. Although N-BEATSx incorporates exogenous variables, its performance drops significantly compared to N-BEATS (e.g., MAPE of 0.635% vs. 52.097% on the US dataset). This raises concerns about the rigor of the experimental process.

[1] TimeXer: Empowering Transformers for Time Series Forecasting with Exogenous Variables. NeurIPS 2024.

**Questions:**

See W1-5.

---

### Note · Authors · 2025-11-27

I have read and agree with the venue's withdrawal policy on behalf of myself and my co-authors.